# LDMX: The Light Dark Matter eXperiment and M³: The Muon Missing Momentum Experiment †

**Matthew Solt** [ID]

Department of Physics, University of Virginia, Charlottesville, VA 22903, USA; gtf9nz@virginia.edu
† Presented at the 23rd International Workshop on Neutrinos from Accelerators, Salt Lake City, UT, USA, 30–31 July 2022.

**Abstract:** The constituents of dark matter are still unknown, and the viable possibilities span a very large mass range. The scenario where dark matter originates from thermal contact with familiar matter in the early Universe requires the DM mass to lie within approximately an MeV to 100 TeV. Considerable experimental attention has been given to exploring weakly interacting massive particles in the upper end of this range (few GeV–TeV), while the MeV to GeV region has been steadily gaining more attention in recent years. If there is an interaction between light DM and ordinary matter, as there must be in the case of a thermal origin, then there is a production mechanism in accelerator-based experiments. The Light Dark Matter eXperiment (LDMX) is a planned electron-beam fixed-target missing-momentum experiment that has unique sensitivity to light DM in the sub-GeV range. Of particular interest to the NuFact muon working group is a proposal for a muon LDMX that uses a muon beam to probe the electron-phobic scenario. This contribution will provide an overview of the theoretical motivation, the main experimental challenges, how they are addressed, and the projected sensitivities in comparison to other experiments.

**Keywords:** dark matter; light dark matter; accelerators; missing momentum; muon physics

## 1. Introduction

There is compelling astrophysical evidence that ∼85% of the matter in the Universe is composed of beyond-standard model (SM) dark matter. While the weakly interacting massive particles (WIMPs) model remains the most popular model of dark matter, stemming from its simplicity in the explanation of dark matter with thermal origins, null results from direct detection and high-energy collider experiments have shifted community interest toward lower mass dark matter in the sub-GeV range, also referred to as dark sectors [1]. Fixed target accelerator-based experiments utilizing a missing momentum technique can probe invisibly decaying mediators that arise from models of sub-GeV thermal dark matter. Two of these future missing momentum experiments, i.e., the Light Dark Matter eXperiment (LDMX) and the muon missing momentum (M³), use an electron beam and a muon beam, respectively, to probe a variety of these models.

## 2. The Light Dark Matter eXperiment

The Light Dark Matter eXperiment (LDMX) is a planned fixed target experiment at SLAC that searches for invisibly decaying mediators using a missing momentum technique [2]. A missing momentum search of this kind would require the tagging and full reconstruction of all initial and final state SM particles. LDMX will use an electron beam of approximately a ∼few GeV, which will be incident on a thin tungsten target to produce mediators. The kinematics of this process result in the mediator (e.g., a dark photon) generally carrying away most of the beam energy; thus, the only remaining visible final state particle is a soft recoil electron. Measuring a large momentum loss between this electron and the initial beam electron is a clear sign of the production of invisible particles, provided that

SM backgrounds are sufficiently vetoed. This mechanism and a schematic of the LDMX apparatus are shown in Figure 1.

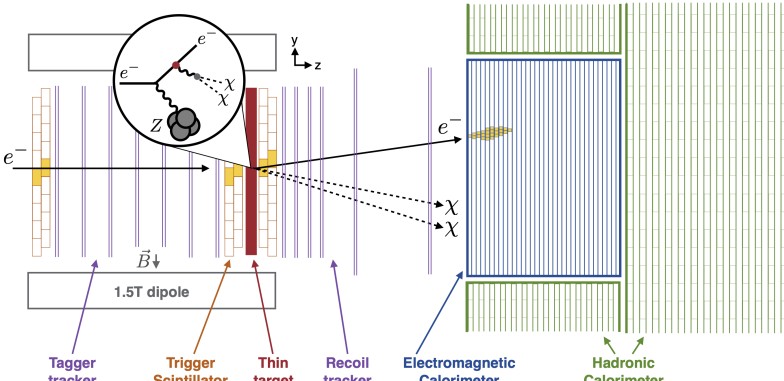

**Figure 1.** A schematic of the LDMX detector. Mediators, such as dark photons, are produced on a thin tungsten target and decay into invisible particles. These mediators generally carry away most of the beam energy; thus, the only visible final state particle is a soft recoil electron. Measuring significant momentum loss between the initial beam electron and the final recoil electron is a clear sign of the production of invisible particles.

The beam will utilize the existing SLAC linear accelerator and will run parasitically with existing experiments. The Linac to End Station A (LESA) is a planned dedicated transfer line that will deliver the beam to LDMX at End Station A. Since the full initial and final states of signal and background processes must be individually tagged and fully reconstructed, the beam structure is designed with a low $e^-$ multiplicity at a high rate of 37 MHz. The current schedule is proposed to run in two phases, with a 4 GeV electron beam with $\sim 10^{14}$ electrons on target for Phase I and an 8 GeV electron beam with $\sim 10^{16}$ electrons on target for Phase II.

The LDMX apparatus is designed to be hermetic with high-momentum resolution and a high veto efficiency of SM backgrounds. The tagging and recoil trackers provide offline measurements of the incoming electron and outgoing recoil electron momentum, respectively. The modules for the trackers use silicon microstrip sensors based on the design of the heavy photon search silicon vertex tracker [3,4]. The current design of the target is a thin $0.1 X_0$ radiation length of the tungsten, which sits between the tagging and recoil trackers. Downstream of the trackers is the electromagnetic calorimeter (Ecal), which provides the missing energy trigger and a veto to electromagnetic particles. It is a highly granular W-Si sampling calorimeter based on the HGCAL upgrade for the CMS experiment [5]. The furthest downstream sub-detector is the hadronic calorimeter (Hcal), which is designed to veto neutral hadrons with high efficiency. It is a segmented array of plastic scintillators with embedded wavelength-shifting fibers and steel absorbers based on the Mu2e Cosmic Ray Veto design [6]. Interspersed in the tagging tracker are the trigger scintillator arrays, which provide a fast count of electron multiplicity for the global trigger.

In order for such a search to be possible, SM backgrounds must be mitigated with high efficiency. The highest background rates include scattered beam electrons, bremsstrahlung, and QED trident production. A simple missing energy trigger with online energy information provided by the Ecal, which retains high signal efficiency due to a single soft recoil electron in the final state, eliminates most of these processes. Next, bremsstrahlung conversions to charged hadrons via photonuclear (PN) processes or conversions to muon pairs (typically occurring in the Ecal) will pass the missing energy trigger and must be vetoed offline. The high-granularity capabilities of the Ecal provide full shower containment, which enables a high-efficiency separation of hadronic showers from PN processes and electromagnetic showers by utilizing information from the respective shower shapes. The high-granularity capabilities of the Ecal also enable tracking of minimum ionizing particles (MIPs) from muons or charged hadrons that leave straight tracks in the Ecal. PN processes

that produce hard neutral hadrons can punch through the Ecal, but are vetoed by a $\sim 16\ \lambda$ deep Hcal. The Hcal also contains a much wider side portion in order to veto wide-angle bremsstrahlung processes that would otherwise miss the detector. Finally, irreducible processes that can mimic a missing momentum signature, such as Moller scattering, accompanied with charged-current quasi-elastic scattering or neutrino production, are expected to occur at a rate much less than $10^{-16}$. Thus, these processes are not a concern at the proposed luminosity.

A detailed simulation involving $\sim 10^{14}$ electrons on target at 4 GeV shows that when all subsystems are combined, less than one background event is expected while retaining high signal efficiency ($\sim 30$–50% across the mass range of interest) [7,8]. This enables sensitivity projections to be made for both phases of running, as shown in Figure 2. These projections show that LDMX is able to improve upon existing constraints by several orders of magnitude and it has sensitivity to the most favorable models of thermal dark matter in the sub-GeV range. Finally, an extra veto handle, which was not used for this background rejection, is the electron transverse momentum distribution, as measured by the recoil tracker, which can be used for further background rejection if needed.

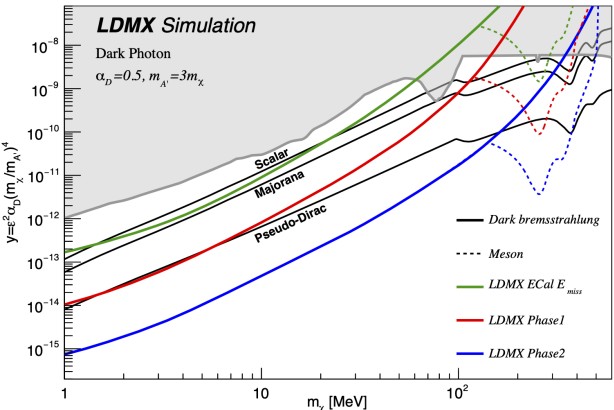

**Figure 2.** The projected sensitivity of the LDMX experiment for Phase I (red) and Phase II (blue), as defined in the text, as well as when using the Ecal as an active target (green). In this figure, $y$ is a dimensionless parameter, $\epsilon$ is the kinetic mixing parameter, $\alpha_d$ is the dark photon$-$dark matter coupling constant, $m_\chi$ is the mass of the dark matter particle, and $m_{A'}$ is the mass of the dark photon [9].

Beyond the standard missing momentum search, LDMX offers a much broader physics program. Since LDMX effectively operates as an active beam dump, it can competitively search for long-lived visibly decaying particles through a sudden appearance of daughter particles within the Ecal or Hcal [10]. Through this technique, LDMX can probe several models of interest to thermal dark matter, including dark photons, axion-like particles (ALPs), and strongly interacting massive particles (SIMPs). In addition, because LDMX fully reconstructs the final and initial states of every event, it can make electro-nuclear measurements that are of interest to neutrino experiments [11].

## 3. The Muon Missing Momentum Experiment

The muon missing momentum ($M^3$) is a proposed experiment that utilizes a muon beam and an LDMX-like apparatus for a missing momentum search for beyond SM particles [12]. A muon beam would enable searches for muon-philic bosons, which are motivated by both the muon g-2 anomaly and models of thermal dark matter, and would otherwise be unobserved with electron beam experiments. Thus, these searches would be complementary. Beyond this, using a muon beam has an advantage over an electron beam in that the bremsstrahlung backgrounds are suppressed by several orders of magnitude $\left(\frac{m_e}{m_\mu}\right)^2$. $M^3$ also has an advantage over similar high-energy muon beam experiments in terms of its compact size and excellent momentum resolution.

To address the two main motivations, the experiment is proposed to operate in two phases. Phase I can probe the entire favored parameter space for the muon g-2 anomaly with a 15 GeV muon beam and $10^{10}$ muons on target. Phase II can probe much of the favored parameter space for sub-GeV thermal dark matter with a 15 GeV muon beam and $10^{13}$ muons on target. Similar to LDMX, the beam would need to have a high repetition rate with up to a few muons per bunch. Viable facilities at Fermilab that could produce such a beam structure are the M-test facility and SpinQuest beamline.

The apparatus for $M^3$ is proposed to be similar to LDMX and is shown in Figure 3. The design contains both a tagging and recoil tracker to measure incoming and outgoing muon momenta, respectively, as well as a measurement of charged-track multiplicity. Downstream of the trackers is an Ecal used for triggering and rejecting electromagnetic backgrounds and an Hcal used to reject neutral hadrons (though not as deep as LDMX since the PN backgrounds are significantly suppressed). The major difference between the designs of these detectors is a thick active target for $M^3$ as opposed to a thin metal target for LDMX. This enables increased signal production in $M^3$ as well as a measure of SM energy deposition in the target to further reject backgrounds that could mimic a missing momentum signature. It is suggested that the LDMX Ecal material be used as the thick active target, though LYSO is also a viable possibility.

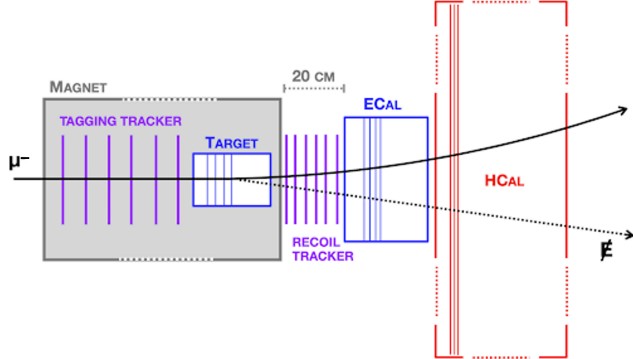

**Figure 3.** The proposed $M^3$ detector, which utilizes an LDMX-like apparatus with a muon beam for a missing momentum search. The major difference between the $M^3$ and LDMX detectors is the thick active target, as opposed to a thick tungsten target.

The SM backgrounds for $M^3$ are similar to LDMX. Large amounts of bremsstrahlung, though at a suppressed rate, create secondaries that must be vetoed. The Ecal and Hcal for $M^3$ operate similarly in terms of vetoing such backgrounds. Additional backgrounds unique to $M^3$ arise from a large spread in the beam momentum and beam contamination from pions. The former can be mitigated by measuring the initial momentum in the tagging tracker while the latter can be mitigated by additional hadronic absorbers (though at the expense of increased beam momentum spread).

A preliminary study based on $10^7$ muons on target, with a few simple selections on energy deposition in the active target, Ecal, and Hcal shows zero background with high signal efficiency. Though not as rigorously studied as the backgrounds in LDMX, it is conceivable that the zero background could be achievable for $10^{13}$ muons on target. With these assumptions, Figure 4 shows the sensitivity for both phases of running for the minimal dark photon model and a muon-philic mediator. Phase I would conclusively probe a muon-philic interpretation of the muon g-2 anomaly. Phase II would complement the LDMX sensitivity for the dark photon model at higher mass and would probe much of the favored parameter space for thermal dark matter for a muon-philic mediator.

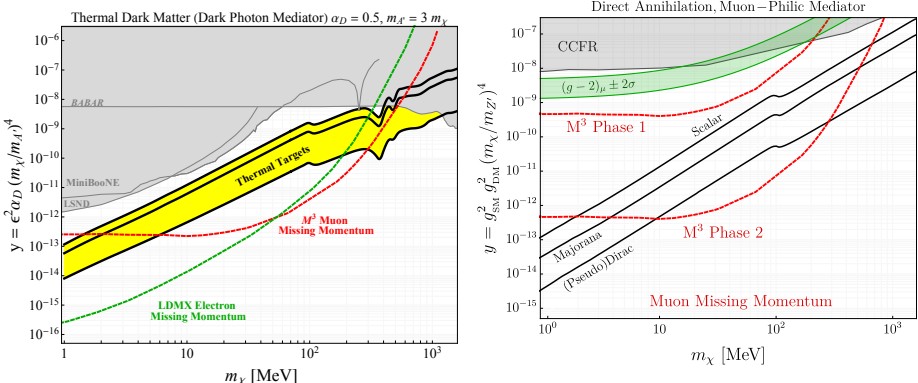

**Figure 4.** The projected sensitivity of M$^3$ for the two proposed phases defined in the text for (**left**) the dark photon model and (**right**) muon-philic mediators. For the muon-philic model, $m_{Z'}$ is the mass of the Z′ and $g_{DM}$ is the gauge coupling of dark matter to the Z′.

## 4. Conclusions

Light dark matter models in the ∼MeV-GeV mass range remain well motivated as an explanation for dark matter with a thermal origin. Many of these models can be probed through accelerator-based missing momentum experiments. The Light Dark Matter eXperiment (LDMX) is a planned fixed target missing momentum experiment that utilizes an electron beam to search for invisibly decaying mediators. Through two phases of running, i.e., a shorter phase consisting of a few months of data followed by a longer ∼3-year phase, LDMX can conclusively probe most of the MeV-GeV parameter space for favored models of thermal dark matter. The muon missing momentum (M$^3$) experiment is a proposed active target missing momentum experiment that utilizes a muon beam and an LDMX-like apparatus to probe muon-philic mediators in the sub-GeV mass range. The first proposed phase can conclusively probe a muon-philic interpretation of the muon g-2 anomaly while the second proposed phase can probe much of the favored parameter space for thermal dark matter with a muon-philic mediator.

**Funding:** The author acknowledges support from an Intensity Frontier base grant DE-SC0007838 from the Office of High Energy Physics in the United States Department of Energy Office of Science.

**Institutional Review Board Statement:** Not applicable.

**Informed Consent Statement:** Not applicable.

**Data Availability Statement:** Not applicable.

**Acknowledgments:** This work was submitted on behalf of the LDMX collaboration and the authors of the M$^3$ proposal.

**Conflicts of Interest:** The author declares no conflict of interest.

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
