# Peer review of "LDMX: The Light Dark Matter eXperiment and M3: The Muon Missing Momentum Experimentâ€"

_psf, doi:10.3390/psf8010017_

Round 1
Reviewer 1 Report
The paper "LDMX: The Light Dark Matter eXperiment" is a nice summary of the LDMX and M3 experiments. The article is well written and concise. I would encourage the authors ti include M3 in the title of the article as it results in a full section of the article.
only a couple of stylistic tweaks: the reviewer suggests to reframe the plots of figure 4 to have the same sizes. Line 123: "bremsstrahlung" exceed the paper margins.
Author Response
The author thanks the reviewer for the positive feedback and suggestions on improving the manuscript. I have made the following simple changes in response to the comments.
I have update the title to “LDMX: The Light Dark Matter eXperiment and $M^3$: The Muon Missing Momentum Experiment”
I have attempted to re-align Fig. 4 by forcing Latex to do what I want. Unfortunately those figures were generated independently without regards to alignment and I don’t have an easy way to regenerate them. I think I have to live with the slight misalignment.
Bremsstrahlung has been fixed.
Reviewer 2 Report
Dear Author,
the paper is a Proceedings type and is brief description of the LDMX experiment. In particular after a short introduction on the motivation to search for Light Dark Matter with electron-beam or muon-beam fixed-target missing-momentum experiment, the author gives a general description of the LDMX set-up and the strategy employed to point out the signal produced by the decay of a Dark mediator. In the final part of the article, the possible muon LDMX experiment based on the use of a muon beam to probe the electron-phobic is also shortly addressed. I recommend the paper for the publication in a Proceedings Series after the revision of some part that can improve the clarity of the paper. Here are my comments:
- Line 5: the sentence “the region MeV to GeV is largely unexplored” is not true; there exists many experiments in the direct detection dark matter field searching for GeV to sub GeV Dark Matter candidate that have reported results so far. This sentence has to be rewritten.
- Line 12: it is written that the paper will give “the status of LDMX experiment” but going throughout the paper I cannot find any status information; I think it would be better to change the word “status” with the word “perspectives”; this change will better clarify to the reader what he/she can expect in the following sections of the article.
- Line 39: it is mentioned the LCLS-II experiment; it apparently doesn’t have any connection with LDMX experiment and it is not clear why it is mentioned. I suggest to remove this mention.
- Line 49: what does “excellent resolution” mean? It would be better to give some estimate of this resolution at the energy of interest; much better would be explain how this resolution can affect the sensitivity of the experiment.
- Figure 2: in the plot some parameters appear without any definition (alpha_D, Mx, MA’); I strongly suggest to define then in the text or in the caption.
- Figure 4: as for Figure 2 I strongly suggest to define the parameters that appear in the axes
- In the Conclusion section I suggest to include a time schedule of the LDMX experiment and an estimation of the time needed to reach the reported sensitivity. It is important for an interested reader to understand the time profile of the experiment also considering that in the article two possible phases of the experiment are mentioned.
Author Response
The author thanks the reviewer for the positive feedback and suggestions on improving the manuscript. I address the specific comments below.
- Line 5: the sentence “the region MeV to GeV is largely unexplored” is not true; there exists many experiments in the direct detection dark matter field searching for GeV to sub GeV Dark Matter candidate that have reported results so far. This sentence has to be rewritten.
The intent was to say it is largely unexplored compared to WIMP DM. It has been re-written for clarity: “has been steadily gaining more experimental attention in recent years.”
- Line 12: it is written that the paper will give “the status of LDMX experiment” but going throughout the paper I cannot find any status information; I think it would be better to change the word “status” with the word “perspectives”; this change will better clarify to the reader what he/she can expect in the following sections of the article.
The status of the LDMX experiment is complicated and has changed quite dramatically even since first submission. This phrase was added more for the actual talk. I have removed the phrase and have added the run requirements for the two phases as suggested below.
- Line 39: it is mentioned the LCLS-II experiment; it apparently doesn’t have any connection with LDMX experiment and it is not clear why it is mentioned. I suggest to remove this mention.
LCLS-II is the current large scale experiment at SLAC. Readers familiar with the facility might be wondering what happens to LCLS-II while LDMX is running. The answer is that it is a shared beam line and runs parasitically only taking a small fraction of the current. I have kept that LDMX runs parasitically, but removed reference to LCLS-II.
- Line 49: what does “excellent resolution” mean? It would be better to give some estimate of this resolution at the energy of interest; much better would be explain how this resolution can affect the sensitivity of the experiment.
I have removed "excellent resolution” and to quote a specific number (or discuss how it affects the sensitivity) is a bit complicated and beyond the scope of these proceedings.
- Figure 2: in the plot some parameters appear without any definition (alpha_D, Mx, MA’); I strongly suggest to define then in the text or in the caption.
Added
- Figure 4: as for Figure 2 I strongly suggest to define the parameters that appear in the axes
Added
- In the Conclusion section I suggest to include a time schedule of the LDMX experiment and an estimation of the time needed to reach the reported sensitivity. It is important for an interested reader to understand the time profile of the experiment also considering that in the article two possible phases of the experiment are mentioned.
A few months for phase I and ~3 years for phase II. It has been added.